# Study on Optimizing Novel Antimicrobial Peptides with Bifunctional Activity to Prevent and Treat Peri-Implant Disease

**DOI:** 10.3390/antibiotics11111482

**Published:** 2022-10-26

**Authors:** Shuipeng Yu, Qian Zhang, Meilin Hu, Borui Zhao, Zhiyang Liu, Changyi Li, Xi Zhang

**Affiliations:** 1School and Hospital of Stomatology, Tianjin Medical University, Tianjin 300070, China; 2College of Electronic Information and Optical Engineering, Nankai University, Tianjin 300071, China

**Keywords:** antimicrobial peptides, macrophage polarization, anti-inflammatory, bone immunoregulatory, osseointegration

## Abstract

The bacterial invasions and inflammatory responses after implant placement often affect osseointegration; the increased secretion of pro-inflammatory cytokines can lead to poor formation of bone and bone absorption. Previous research has shown that the antimicrobial peptide 6K-F17 has antibacterial and immunomodulatory properties. The objective of this study was to optimize KR−1 and KR−2, based on 6K-F17, to apply to the tissue around the oral implant. Our first objective is to study its antibacterial properties, and then we intend to further study its osteogenic ability to osteoblasts by modulating the immune response of macrophages. In this research, KR−1 and KR−2 can inhibit the formation of bacterial biofilm, and further kill bacteria *S. gordonii* and *F. nucleatum* by destroying the cell wall and cell membrane of bacteria. The novel peptides restrained the activation of the NF-κB signaling pathway by reducing the phosphorylation levels of IκBα and p65, inhibiting the degradation of IκBα and the nuclear translocation of p65, and increasing the percentage of M2 phenotype in macrophages. This suppressed the inflammatory response induced by lipopolysaccharides and enhanced the osteogenic activity of osteoblasts; this, in turn, promoted osteogenesis. The antimicrobial peptide KR−1 showed better performance. Our results demonstrate that KR−1 and KR−2 have antibacterial and bone immunomodulatory effects, and further promote osteogenesis by modulating the immune microenvironment, which provides the possibility for the adjuvant treatment of peri-implant diseases.

## 1. Introduction

The emergence of implants has significantly changed previous repair concepts. Implants have become the first choice for patients with missing teeth because they can restore masticatory function and provide aesthetic benefits without damaging the natural teeth [1]. Implant materials used in clinics are mainly pure titanium and titanium alloys [2], which have good biological properties [3] and osseointegration abilities [4]. However, a layer of titanium dioxide (TiO_2_) is rapidly formed on the surface after titanium is exposed to air or liquid. This reduces the biological activity of the titanium base, rendering it incapable of stimulating the proliferation of osteoblasts. Consequently, osseointegration is delayed, and its quality is reduced, which may lead to implant failure [5,6]. It often takes several months to achieve osseointegration after implant placement, and rapid osseointegration is important for the success of implant surgery [7].

In the early stage of implant placement, bacteria colonize the implant surface and proliferate, thereby inducing the body’s local immune response. Excessive bacterial proliferation and inflammatory response will affect the early osseointegration of implants and lead to implant failure. The application of drugs with antibacterial and immunomodulatory activity around implants will provide a new attempt to prevent and treat peri-implant diseases. Macrophages are the main effector cells of the immune response [8]. Under the stimulation of different signaling factors, macrophages can be further differentiated into two types: pro-inflammatory macrophages (M1) and anti-inflammatory macrophages (M2). The differentiation results of different directions affect the immunomodulation of macrophages during tissue repair [9].

In recent years, antimicrobial peptides have been used in different disciplines, such as orthopedics [10,11], cancer [12,13], ophthalmology [14], dermatology [15], and stomatology [16,17], because of their excellent antibacterial [18], antiviral [19], anticancer [20] and immunomodulatory [21] characteristics; they are considered to be the most promising drugs [22] in terms of solving the global problem of antibacterial drug resistance. A membrane-active peptide, 6K-F17 (sequence: KKKKKKAAFAAWAAFAA), has been found, and it is different from the natural cationic antimicrobial peptide with an amphiphilic design. Its positively charged residues are located on one surface of the helix, and six Lys residues gather at the N-terminal of the sequence [23]. The 6K-F17 peptide is non-toxic to human bronchial epithelial cells [24] and can downregulate the pro-inflammatory cytokines interleukin-6 (IL-6), interleukin-8 (IL-8), and tumor necrosis factor-α (TNF-α) infected human bronchial epithelial cells through its immunomodulation [25]. Several studies have found that amphiphilicity is an important characteristic of antimicrobial peptides for the invasion of bacterial and cell membranes; it directly affects the mechanism of action of antimicrobial peptides [26]. Therefore, it is important to adjust the sequence of 6K-F17 to optimize the antibacterial and anti-inflammatory performance for oral application.

Based on the above, this study modified the antimicrobial peptide 6K-F17. The special structure of the Lys residue aggregation was preserved, and the uncharged amino acid at the other end was replaced to obtain two new antimicrobial peptides: KR−1 (KKKKKKRAFARWRAFAR) and KR−2 (KKKKKKRRFRRWRRFRR). The basic properties and structures of the two new antimicrobial peptides were studied, and their cytotoxicity was determined. The antibacterial activity of KR−1 and KR−2 were evaluated against *Streptococcus gordonii* and *Fusobacterium nucleatum*. The effects of the peptides on macrophage polarization and osteogenic ability via the regulation of macrophage immune response were studied. We hope that the new antimicrobial peptides, KR−1 and KR−2, can inhibit the formation of oral biofilm and adjust the immune regulation of macrophages, providing the possibility for the prevention and treatment of peri-implant diseases.

## 2. Results

### 2.1. Properties of Antimicrobial Peptides

To study the structure of KR−1 and KR−2, their theoretical, physical and chemical properties were predicted using the peptide calculator. The results are shown in Figure 1A,B. Both antimicrobial peptides demonstrated excellent solubility. The hydrophilic group of KR−1 accounts for 58% of all amino acids; the hydrophilic group of KR−2 accounts for 82%. Both have a net positive charge; the net positive charge of KR−1 is +10, while that of KR−2 is +14. KR−1 has an alternating arrangement of hydrophilic and hydrophobic fragments, while KR−2 is composed of hydrophilic amino acids and aromatic amino acids.

Alphafold was used to predict the tertiary structures of two new peptide chains with different compositions. The tertiary structure of each peptide was predicted five times, based on the internal confidence of the software. The three-dimensional structure prediction diagram shows that KR−2 is only a chain structure, while KR−1 has a better spiral structure and better stability.

### 2.2. Antimicrobial Activity against S. gordonii and F. nucleatum

The Minimum Inhibitory Concentration (MIC) and Minimum Bactericidal Concentration (MBC) of the two antimicrobial peptides against *S. gordonii* and *F. nucleatum* were determined (Table 1). For *S. gordonii*: MIC of KR−1 is 125 μg/mL, MBC is 250 μg/mL; MIC of KR−2 is 50 μg/mL, MBC is 150 μg/mL; for *F. nucleatum*: MIC of KR−1 is 100 μg/mL, MBC is 200 μg/mL; MIC of KR−2 is 50 μg/mL, MBC is 100 μg/mL. Compared with KR−1, KR−2 showed stronger antibacterial activity.

### 2.3. Biofilm Inhibition

The effects of antimicrobial peptides on the biofilm formation ability of *S. gordonii* and *F. nucleatum* were observed according to the biofilm susceptibility test. The OD values of the two bacterial biofilms treated with antimicrobial peptides KR−1 and KR−2 were significantly decreased (Figure 2A), and the difference was statistically significant.

AO/EB staining can observe the bactericidal ability of antimicrobial peptides and the ability to destroy biofilms. Figure 2B shows that the control group is mainly green (living bacteria), while the bacteria treated with the two antimicrobial peptides are mainly red (dead bacteria), indicating that both antimicrobial peptides KR−1 and KR−2 can significantly kill bacteria. Among them, the bacterial biofilm treated with KR−2 showed significant destruction compared with the control group.

### 2.4. Scanning Electron Microscope (SEM)

A scanning electron microscope (SEM) can more intuitively observe the effects of antimicrobial peptides KR−1 and KR−2 on the morphology of *S. gordonii* and *F. nucleatum*. As shown in Figure 3A, both *S. gordonii* and *F. nucleatum* in the control group had complete bacterial morphological structures. The treatment of KR−1 and KR−2 can significantly destroy the bacterial morphology of *S. gordonii*, rupturing the cell wall and cell membrane to kill the bacteria; at the same time, KR−1 can significantly shrink and deform the cell wall of *F. nucleatum* and further lead to death, while KR−2 has a stronger effect on *F. nucleatum* than KR−1.

### 2.5. Biocompatibility of Antimicrobial Peptides

RAW 264.7 was used to study the biocompatibilities of KR−1 and KR−2. Based on the OD value of CCK-8 (Figure 3B,C), KR−1 showed no cytotoxicity at concentrations less than 256 μg/mL, but it promoted cell proliferation better at a concentration of 64 μg/mL. In contrast, KR−2 had poor biocompatibility. At low concentrations below 16 μg/mL, it had no cytotoxic effect on cells and could also significantly promote cell proliferation, but its cytotoxicity increased sharply with the increase in concentration. The above results show that antimicrobial peptides have good biocompatibility at specific concentrations. Therefore, the antimicrobial peptides concentrations used in this study were as follows: KR−1:64 μg/mL and KR−2:16 μg/mL.

### 2.6. Effect of Peptides on the Expression of Pro-Inflammatory and Anti-Inflammatory Genes of RAW 264.7

Figure 4 shows the expressions of pro-inflammatory and anti-inflammatory genes under inflammatory conditions. Compared with BLANK group, we added LPS to stimulate macrophages to differentiate into M1 phenotype; on days 1 and 3, the addition of LPS significantly increased the expressions of all the pro-inflammatory genes *CD80*, *iNOS*, *IL1β*, and *TNF-α*. The expressions of pro-inflammatory genes decreased by varying degrees after the addition of different antimicrobial peptides, and the decrease effect of the KR−1 group was much bigger than the KR−2 group (Figure 4A–D).

The expressions of anti-inflammatory genes CD206, CD163, and Arg1 were downregulated on days 1 and 3 after the addition of LPSs and were more significant on day 3 (Figure 4E–G). Compared to the LPS group, the downregulation of gene expression was alleviated after the addition of different antimicrobial peptides. The ability to reverse the decline of anti-inflammatory gene expression was prominent on day 3 in the KR−1 group.

### 2.7. Effects of Antimicrobial Peptides on Polarized Morphology of RAW264.7

Macrophages polarize in different directions, and they have significantly different cell morphology [27,28]. As shown in Figure 5a–d,A–D, on day 1, the nucleus in the BLANK group (Figure 5a) was evident, and the cytoskeleton was next to the nucleus, maintaining its original shape. After the addition of LPS (Figure 5b), the cytoskeleton of most cells began to expand circularly around the nucleus. The mode of expansion of the cytoskeleton changed significantly in the LPS + KR−1 (Figure 5c) and LPS + KR−2 (Figure 5d); they tended to elongate.

On day 3, the cytoskeleton in the BLANK group (Figure 5A) expanded, although it maintained its original morphology without significant changes. In the LPS group (Figure 5B), an obvious circular diffusion of the cytoskeleton was observed. The cytoskeleton changed to a slender shape under inflammatory conditions after adding KR−1 (Figure 5C), and the slender cells increased significantly. This kind of morphological change was also observed in the KR2 + LPS group (Figure 5D), but it was less than that in the KR−1 group.

Flow cytometry (Figure 5E) showed that the percentage of CD86^+^ (M1 marker) cells in the LPS group was 19.6%, down-regulated to 13.1% in the LPS + KR−1 group, and down-regulated to 15.8% in the LPS + KR−2 group. Meanwhile, the percentage of CD206^+^ (M2 marker) cells in LPS + KR−1 group was up-regulated to 30.8%, which was much higher than that in LPS group (7.61%) and PBS group (16.6%).

### 2.8. Antimicrobial Peptides Pair Regulation of NF-κB-p65 Signal Pathway during RAW 264.7 Polarization

Western blot experiments were used to determine whether the classical or non-classical NF-κB signaling pathway was activated during the antimicrobial peptide regulation of macrophage polarization. The experimental results (Figure 6A) showed that the presence of the peptide significantly reduced IκBα protein phosphorylation and degradation. Simultaneously, the level of p65 phosphorylation increased after the induction of LPS, while the presence of antimicrobial peptides significantly inhibited p65 phosphorylation.

To further explore the potential mechanism, p65 immunofluorescence staining was performed for different groups of cells, and the outcomes are depicted in Figure 6B. After the induction of LPS, p65 was translocated from the cytoplasm to the nucleus and mainly concentrated in the nucleus. In the KR−1 and KR−2 groups, p65 mainly existed in the cytoplasm; it significantly inhibited the translocation of p65 to the nucleus, and the inhibitory effect of KR−1 was more pronounced.

### 2.9. Regulation of Conditioned Medium on Osteogenic Ability of Osteoblasts MC3T3-E1

To assess the immunomodulatory functions of KR−1 and KR−2 on osteoblasts, MC3T3-E1 cells were grown in an osteogenic medium containing CMBLANK, CMLPS, CMLPS + KR−1, and CMLPS + KR−2, respectively. ALP staining after 7 days of culture (Figure 7A) showed that the LPS group had decreased ALP activity. However, after adding antimicrobial peptides, ALP activity increased significantly, and the ALP level of the KR−1 group was higher than that of the BLANK group. The same result was obtained for the expression analysis of day 7 osteogenesis-related genes ALP, Runx-2, and COL1α1, which belaved an increase in the expression of osteogenic genes (Figure 7B–D).

ARS staining was performed on the cells from each group on day 14. The results are depicted in Figure 7E. Alizarin red staining was deepest for the CMLPS + KR−1 group. The staining of the CMLPS + KR−2 and control CMBLANK groups were similar, but almost no staining was observed for the CMLPS group. Similarly, osteogenesis-related genes OCN, Runx-2, and COL1α1 were tested on day 14, and the addition of LPS reduced the expression of osteogenic genes, while the peptide reversed the inhibitory effect of LPS on osteogenic gene expression. The gene expression level of the CMLPS + KR−1 group was higher than CMBLANK group (Figure 7F–H), and the levels of expression of osteogenesis-related proteins were detected using Western blotting, the outcomes illustrated that they were consistent with the gene expressions (Figure 7I).

## 3. Discussion

Implant repair has been widely used in oral repair of patients with dentition defects and deletions in recent years, on account of its advantages [29]. However, several factors, including inflammation, can lead to implant failure [30]. Some studies see bacteria colonize around the implant to form a biofilm community [31], achieve symbiotic balance with the host, and not affect implant osseointegration during the early stages after implantation. However, the factors promoting biofilm growth are also conducive to tissue inflammation [32]. The occurrence of peri-implant inflammation promotes osteolysis, inhibits new bone formation, and affects osseointegration during the early stages after implantation, resulting in poor bone formation and implant failure [33,34]. During local inflammation, macrophages differentiate to the M1 phenotype and secrete pro-inflammatory factors that affect the osteogenic activity of osteoblasts [35]. Researchers have found that the pro-inflammatory cytokines IL-1β and TNF-α can affect the activity of osteoblasts and promote bone resorption [34,36].

Some scholars have proposed that the surface of inert titanium implants can be biologically modified to improve the immune microenvironment around the implant [37,38]. Some studies on the biological modification of implant surfaces have used antimicrobial peptides to promote osteogenesis [39]. Antimicrobial peptides have an almost unlimited sequence space, which makes them promising for the prevention and treatment of several diseases [40]. The peptide 6K-F17, with its special structure, has been proven to have very strong antibacterial properties [25]. Simultaneously, arginine residues in NF-κB play a significant role in activation and LPS, which can affect the cellular immune response [41,42]. In this study, arginine was used to replace the hydrophobic amino acid at the C-terminus of 6K-F17 to reduce the cytotoxicity of its hydrophobic long chain in mammalian cells. Two new antimicrobial peptides, KR−1 and KR−2, were synthesized: their antibacterial properties—biocompatibility, immunomodulatory performance, and promotion the ability of osteoblasts to form bone—were systematically studied.

Small changes in amino acid composition can change the overall geometric structure of the antimicrobial peptide [43]. In this study, the compositions of the two peptide chains were similar; however, their secondary structures were different. KR−1 had a more helical structure, which was more conducive to its function [44]. Additionally, KR−1 was less cytotoxic than KR−2, indicating that KR−1 had better biocompatibility, which may be related to the different quantity of amino acid substitutions at the C-termini of the two peptide chains and the consequent different hydrophilic hydrophobic ratios [45]. The optimal range of the percentage of hydrophobic residues in the peptide chain is approximately 40–60%, and high percentages may increase cytotoxicity and reduce stability [46]. This is identical with the results of this research.

Previous studies have shown that antimicrobial peptides should have at least one cationic part and one hydrophobic part, which are more conducive to their specific functions. In this study, the antimicrobial peptide 6K-F17 was modified, and the antibacterial capabilities of two new antimicrobial peptides KR−1 and KR−2 are studied. The results of this study show that two novel antimicrobial peptides have significant killing effects on *S. gordonii* and *F. nucleatum*. Compared with KR−1, the antibacterial ability of KR−2 is better, which may be due to the different bactericidal mechanisms of antimicrobial peptides with different structures.

It has been confirmed that LPS can induce inflammatory bone loss [47], and LPS is recognized as a bacterial polysaccharide that can induce macrophages to differentiate into M1 phenotype and secrete pro-inflammatory factors [48]. In this study, LPS was used to induce the differentiation of macrophages into the M1 phenotype. The inflammatory microenvironment was successfully simulated. Afterwards, antimicrobial peptides KR−1 and KR−2 were added. These regulate the tendency of macrophages to polarize towards the M2 phenotype, inhibit the production of pro-inflammatory cytokines, and suppress the inflammatory osteolytic response, which can provide osteoblasts with a more conducive immune microenvironment to promote osseointegration. This is consistent with the research conclusions of Wang [49]. Macrophage showed different cell polarization morphology after it was activated by LPS, and antimicrobial peptides were added to it. Chen et al. [27] reported that changes in cell morphology generally indicate the direction of cell differentiation. When the macrophage M1 phenotype polarized, the cells tended to be round, and when macrophages differentiated to the M2 phenotype, the cells were slender. Chen et al. [28] came to the identical conclusion. In this research, the results also confirmed that macrophages differentiated in different directions, and that their cell morphology was inconsistent.

NF-κB is a significant nuclear transcription factor in cells and plays a key role in many processes, such as inflammatory and immune responses [50,51]; therefore, inhibiting the NF-κB signal transduction pathway through drugs may become a means of treatment. Studies have confirmed that the inhibition of the NF-κB signaling pathway can promote the differentiation of macrophages to the M2 phenotype, secrete anti-inflammatory factors, and inhibit inflammatory bone destruction [52,53]. The p65 is a member of the NF-κB family, and participates in the process of the classical signal pathway activation [54]. The experimental results showed that the peptides inhibit the activation of the NF-κB signaling pathway by inhibiting p65 phosphorylation, which can reduce the M1 phenotype differentiation of macrophages and secretion of pro-inflammatory factors, promote the differentiation of macrophages into the M2 phenotype, and effectively promotes bone formation in inflammatory conditions.

## 4. Materials and Methods

### 4.1. Synthesis and Characterization of Antimicrobial Peptides

The new antimicrobial peptides KR−1 and KR−2 were synthesized by Dechi Biosciences (Shanghai, China). The final peptide sequences were purified by HPLC (>95% purity) and confirmed by mass spectrometry. The basic properties and secondary structure information of the KR−1 and KR−2 sequences were obtained using a peptide property calculator (http://www.pepcalc.com/, accessed on 5 April 2021). The tertiary structures of the two peptide chains were predicted using Alphafold 2.1 software. The peptide was dissolved in a 1×phosphate-buffered solution (PBS, Solarbio, Beijing, China) to prepare high-concentration stock solutions (5 mg/mL) and peptide dilutions of different concentrations.

### 4.2. Behaviors of S. gordonii and F. nucleatum Cultured with the Different Antimicrobial Peptides

#### 4.2.1. Minimum Inhibitory Concentration (MIC) and Minimum Bactericidal Concentration (MBC)

MIC is the lowest concentration at which turbidity can be seen in the medium, while MBC is the lowest concentration at which no bacterial growth is seen on agar medium. *S. gordonii* (No 10558, ATCC) was cultured on brain–heart infusion (BHI) agar plates, *F. nucleatum* (No 25586, ATCC) was cultured in completely anaerobic fashion on CDC anaerobic blood agar plates, and individual colonies were picked and cultured in BHI liquid medium (*S. gordonii* cultured for 24 h, *F. nucleatum* cultured for 48 h).

A specific concentration gradient was set in the range of 500–0 μg/mL. Different concentrations of antimicrobial peptides KR−1, KR−2 solution and BHI bacterial suspension (1 × 10^6^ CFU/mL) were mixed in a 1:1 ratio and inoculated in a 96-well plate, respectively. This was done so that the drug concentration would reach a gradient concentration, and the final volume would be 200 μL. Consequently, the control group was added with sterile BHI medium. Optical density at 600 nm was measured by a microplate reader (Cytation 5; Bio-Tek, VT, Winooski, USA) to determine bacterial growth after anaerobic incubation of the mixture for 24 h (*S. gordonii*)/48 h (*F. nucleatum*). A volume of 10 μL of the liquid in the MIC assay well plate was spread evenly onto blood agar plates and incubated at 37 °C for 48 h. The final concentration corresponding to no bacterial growth is called MBC.

#### 4.2.2. Biofilm Susceptibility Assay

Crystal violet staining was used to evaluate the effects of KR−1 and KR−2 on the biofilm formation of bacteria *S. gordonii* and *F. nucleatum*, respectively. The corresponding MIC concentrations of KR−1 and KR−2 were co-cultured with *S. gordonii* and *F. nucleatum* (1 × 10^6^ CFU/mL) in 96-well plates for 24 h (*S. gordonii*)/48 h (*F. nucleatum*), respectively. They were fixed with 95% methanol, stained with 0.5% (*w/v*) crystal violet (Yuanye Bio-Technology, Shanghai, China), rinsed with PBS and destained with ethanol, and had absorbance measured at 600 nm.

#### 4.2.3. Confocal Laser Scanning Microscopy (CLSM)

The killing effect of antimicrobial peptides on bacteria *S. gordonii* and *F. nucleatum* was observed by CLSM. The bacterial suspension (1 × 10^6^ CFU/mL) was inoculated in a confocal dish for 24 h (*S. gordonii*)/48 h (*F. nucleatum*), and the antimicrobial peptides KR−1, KR−2 and PBS were added to the culture medium, respectively. The final drug concentration reached the MIC concentration, and after co-incubating for 24 h, the cells were washed with PBS, stained with acridine orange/ethidium bromide (AO/EB) (Yuanye Bio-Technology, Shanghai, China), and observed under a laser confocal microscope (Olympus FV1000, Tokyo, Japan).

#### 4.2.4. Scanning Electron Microscopy (SEM)

The effect of antimicrobial peptides on the morphology of bacteria *S. gordonii* and *F. nucleatum* was observed under SEM. The bacterial suspension of 1 × 10^6^ CFU/mL was cultured in a 5 mL centrifuge tube for 24 h (*S. gordonii*)/48 h (*F. nucleatum*), respectively. Afterwards, KR−1 and KR−2 at the final concentration of MIC were added to the cells to continue culturing for 24 h. Centrifugation was performed to obtain bacterial pellet; 2.5% glutaraldehyde resuspended the pellet. This was fixed at 4 °C for 2 h, centrifuged, followed by gradient dehydration of ethanol solution (30%, 50%, 70%, 80%, 90%, 95%, 100%, 15 min for each concentration). It was then freeze-dried, sprayed with gold, and observed by a scanning electron microscope (Gemini 300, Zeiss, Germany).

### 4.3. Behaviors of Macrophages RAW 264.7 Cultured with the Different Antimicrobial Peptides

#### 4.3.1. Cell Culture

RAW 264.7 cells (American type Culture Collection, ATCC, VA, USA) were grown in Dulbecco’s modified Eagle’s medium (DMEM; Gibco, Grand Isle, NY, USA) increased with 10% FBS and 1% (*v/v*) penicillin/streptomycin (Gibco, Grand Isle, NY, USA) in a CO_2_ incubator (Thermo Fisher Scientific, Waltham, MA, USA) at 37 °C. The cells were inoculated into 24-well plates, of a density of 3 × 10^4^ cells per well. Except in the case of the BLANK group, peptides were added (final concentration KR−1 64 μg/mL, KR−2 16 μg/mL) and/or lipopolysaccharide (LPS, 1 µg/mL; Sigma Aldrich, MO, USA) after 24 h, and the time point was considered to be day 0. The experiment was divided into six groups: BLANK, LPS (inflammatory model), KR−1, LPS + KR−1, KR−2, and LPS + KR−2 groups.

#### 4.3.2. Cell Proliferation Assay

RAW 264.7 cells were inoculated in 96-well plates. The same final volume of cell culture medium in each well was ensured, before being incubate for 3 days in a gradient of antimicrobial peptide concentrations of 0, 2, 4, 8, 16, 32, 64, 128, 256 and 512 μg/mL. Cell proliferation was detected using CCK-8 (NCM biotechnology company, Zhejiang, China); OD value was measured using a microplate reader (Cytation 5; Bio-Tek, Winooski, VT, USA) at 450 nm, to determine the peptides concentration in subsequent experiments.

#### 4.3.3. Gene Expression of Cell Polarization

To evaluate the effects of the new antimicrobial peptides KR−1 and KR−2 on macrophage polarization under inflammatory conditions, a quantitative real-time polymerase chain reaction (RT-PCR) was performed. On days 1 and 3, RNA extraction was performed for each group of cells using the Trizol reagent (Thermo Fisher Scientific, Waltham, MA, USA). Mouse mRNA-encoding genes for CD80, inducible nitric oxide synthase (iNOS), interleukin-1β (IL1β), TNF-α, arginase-1 (Arg1), CD163, and CD206 were chosen, and the internal reference gene glyceraldehyde 3-phosphate dehydrogenase was used as a control. Using the 2^−ΔΔCt^ method to calculate the results. The primers of the target gene and internal reference gene are shown in Table 2.

#### 4.3.4. Cell Morphology and p65 Immunofluorescence Staining

Cell morphology and p65 transport into the nucleus were observed. RAW 267.4 cells were planted at a density of 1 × 10^5^ cells on square coverslips (24 mm × 24 mm) in each 6-well plate. The culture conditions were the same as those of the BLANK, LPS, LPS + KR−1, and LPS + KR−2 groups during RAW 264.7 macrophage culture. On day 1 and day 3, the cells were placed with 4% paraformaldehyde, and their membranes burst for 3–5 min with 0.25 percent Triton X-100. Subsequently, each group was blocked for 30 min with 1% bovine serum albumin. The p65 was stained with p65 antibody (Abcam, Cambridge, UK) and goat anti-mouse IgG secondary antibody (Alexa Fluor 488). TRITC-rhodamine and DAPI (Thermo Fisher Scientific, Waltham, MA, USA) were used to stain the cytoskeleton and nucleus, respectively. Confocal laser scanning microscopy (CLSM, Zeiss, Baden Württemberg, Germany) was used to capture images under different light sources.

#### 4.3.5. Flow Cytometry

The CD86^+^ and CD206^+^ macrophages were detected by flow cytometry. After RAW264.7 cells were cultured for 3 days, macrophages were collected, incubated with phycoerythrin (PE)-conjugated anti-mouse CD206 and PerCP-conjugated anti-mouse CD86 antibodies (Biolegend) for 30 min in the dark, and then washed with PBS to stain cells. They were further incubated with allophycocyanin (APC)-conjugated F4/80 antibodis for 30 min at 4 °C, washed with PBS and detected by flow cytometry (Becton,Dickinson and Company, BD, Franklin Lakes, NJ, USA). Data were analyzed by Flow jo 10.

### 4.4. Behavior of MC3T3-E1 Cells in Different Antimicrobial Peptides Conditioned Media (CM)

#### 4.4.1. Preparation of Conditioned Medium

RAW 264.7 cells were planted at a density of 3 × 10^4^ cells per well in a 24-well plate and incubated for 24 h. PBS (BLANK group), LPS (1 μg/mL), LPS + KR−1 (64 μg/mL), and LPS + KR−2 (16 μg/mL) were added to each group and incubated for 3 days. The culture medium of each group was centrifuged at 1500 rpm for 15 min, and the supernatant was collected and filtered through a 0.22-μm filter. The obtained supernatant was blended 1:1 with the culture medium (DMEM containing 10% FBS and 1% penicillin/streptomycin) to obtain the conditioned medium (CM). CM from diverse sources was separated into four groups, namely CMBLANK, CMLPS, CMLPS + KR−1, and CMLPS + KR−2, for further experiments.

#### 4.4.2. Cell Culture

MC3T3-E1 cells (CRL-2593; American Type Culture Collection) were inoculated on 24-well plates. When the cell density reached 80%, the cells were supplemented with osteogenic differentiation components, ascorbic acid (50 µg/mL), and β-disodium glycerophosphate (10 mm) and incubated with CM for 7 and 14 days.

#### 4.4.3. Osteogenic-Related Gene Expression

Trizol was used to separate RNA from experimental cells on days 7 and 14, and the expression levels of alkaline phosphatase (ALP), runt-related transcription factor 2 (Runx-2), collagen type I α1 (COL1α1) and osteocalcin (OCN) were determined. Glyceraldehyde 3-phosphate dehydrogenase was used as the internal control. Reverse transcription and real-time polymerase chain reaction were performed, and the 2^−ΔΔCt^ method was used to determine the results. The primers of target gene and internal reference gene are shown in Table 2.

#### 4.4.4. Alkaline Phosphatase Staining Assay

After 7 and 14 days of cultivation, each group of cells was fixed with 4% paraformaldehyde and stained with ALP staining solution, and photographs of each group of cells were recorded with a digital camera (Canon, Tokyo, Japan).

#### 4.4.5. Alizarin Red S (ARS) Staining

After 14 days of cell incubation, the cells were fixed with 95% alcohol for 10 min and stained with 2% Alizarin Red S solution with a pH of 4.2 for 15 min. A digital camera was used to take photographs.

#### 4.4.6. Western Blot Analysis

The RAW 264.7 cells were cultured with PBS, LPS, KR−1, and KR−2 media for 30 min, and MC3T3-E1 cells were grown in various CM osteogenic media for 14 days. Subsequently, the cells were lysed for 30 min in lysis buffer with protease inhibitors (MedChemExpress), and cell lysates were analyzed by gel electrophoresis after samples were collected for processing and transferred to nitrocellulose membranes. The membrane was incubated with the relevant antibody for 12 h at 4 °C before exposure, after blocking had been performed with a 5% bovine serum albumin solution (Solarbio) for 1 h. The following antibodies were used for Western blotting: p65, phospho-p65, IκBα, phospho-IκBα, GAPDH, β-tubulin, *OCN*, *Runx-2*, and *COL1α1* (all 1:1000 dilutions, Abcam, Cambridge, UK).

### 4.5. Statistical Analysis

The data were analyzed with GraphPad Prism 8.4.2 program (USA) and are expressed as mean ± standard deviation (SD). An independent sample *t* test and one-way analysis of variance (ANOVA) were used for analysis. The pre-experimental design was used to determine all inclusion and exclusion criteria, and *p* < 0.05 denoted statistical significance (* *p* < 0.05, ** *p* < 0.01, *** *p* < 0.001, and **** *p* < 0.0001).

## 5. Conclusions

In this study, we designed key components that exert biological activity effects— antimicrobial peptides KR−1 and KR−2. The antibacterial properties of KR−2 are more prominent, but highly cytotoxicity. KR−1 better balances antibacterial properties and biocompatibility, stronger immune regulation, and a better ability to promote bone tissue formation. In conclusion, the antimicrobial peptides inhibited the activation of the NF-κB signaling pathway by preventing the nuclear translocation of p65, reducing the level of polarization of macrophages to the M1 phenotype, and inhibiting inflammatory response. By regulating the immune microenvironment, we can improve the reduction of osteoblast osteogenic ability caused by an inflammatory response to improve the ability of bone formation. This strategy of using antimicrobial peptides for immune regulation to promote bone formation is feasible, convenient, economic, and effective. Antimicrobial peptides can be prepared in different forms and applied to various needs, including implant surface modification, and they have very broad application prospects.

## Figures and Tables

**Figure 1 antibiotics-11-01482-f001:**
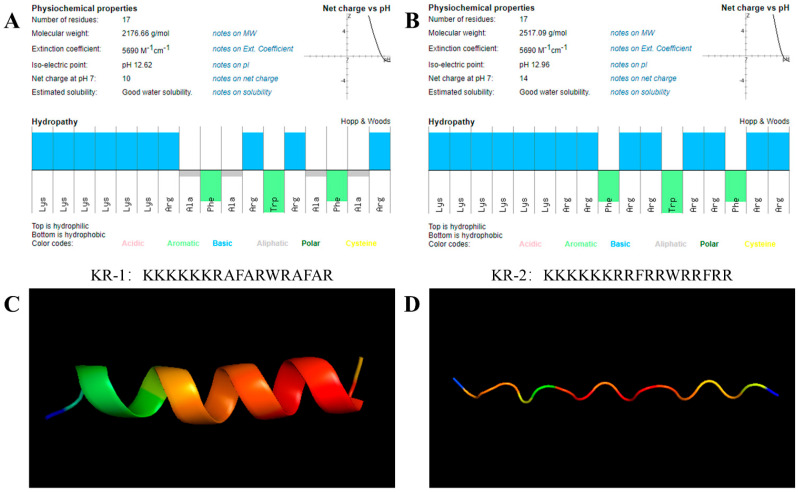
Basic properties of antibacterial peptides. (**A**,**B**) Molecular characteristics and physicochemical properties of amphiphilic antimicrobial peptides KR−1 and KR−2 (top: hydrophilic; bottom: hydrophobic). (**C**,**D**) Alphafold predicted the tertiary structures of KR−1 and KR−2.

**Figure 2 antibiotics-11-01482-f002:**
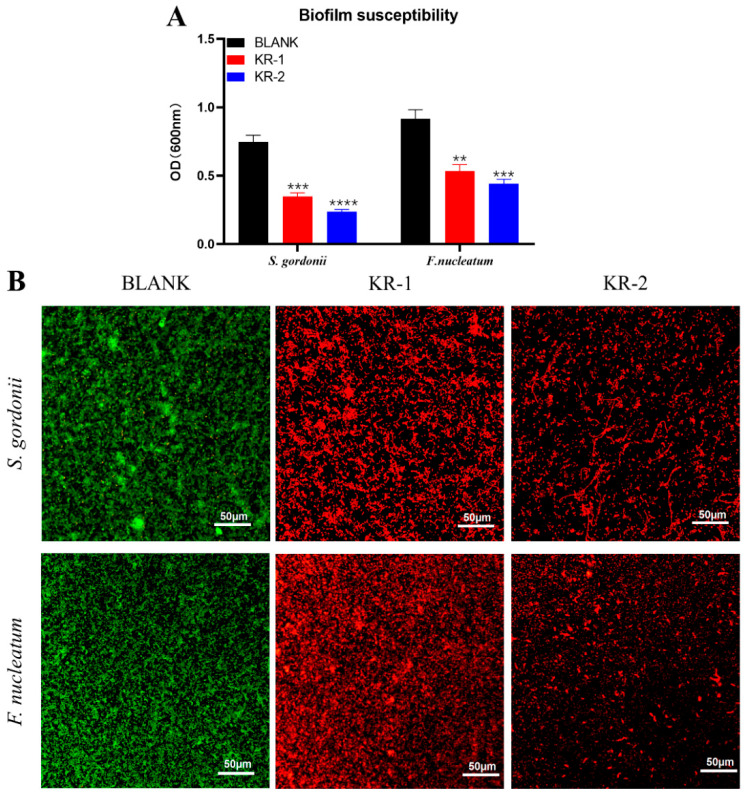
Antibacterial effects of KR−1 and KR−2 on *S. gordonii* and *F. nucleatum*, respectively. Crystal violet-stained OD values (**A**) and AO/EB-stained confocal laser (CLSM) images (**B**) of bacteria treated with different antimicrobial peptides for 24 h (*S. gordonii*)/48 h (*F. nucleatumi*). Note: green represents live bacteria, red represents dead bacteria; data are presented as mean ± standard deviation (*n* = 3). * Indicates statistical significance between the BLANK group and other groups. (** *p* < 0.01, *** *p* < 0.001, and **** *p* < 0.0001, ANOVA).

**Figure 3 antibiotics-11-01482-f003:**
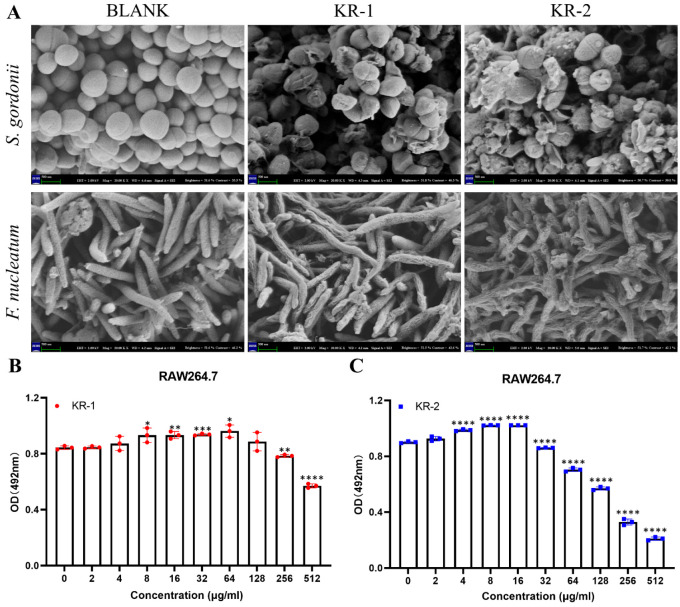
(**A**) *S. gordonii* and *F. nucleatum* were treated with KR−1 and KR−2, respectively, and the changes in bacterial morphology were observed under scanning electron microscope. (**B**,**C**) CCK-8 results of RAW 264.7 when KR−1 and KR−2 were added to the culture for 3 days. Note: data are presented as mean ±standard deviation (*n* = 3). * Indicates statistical significance between the BLANK group and other groups. (* *p* < 0.05, ** *p* < 0.01, *** *p* < 0.001, and **** *p* < 0.0001, ANOVA).

**Figure 4 antibiotics-11-01482-f004:**
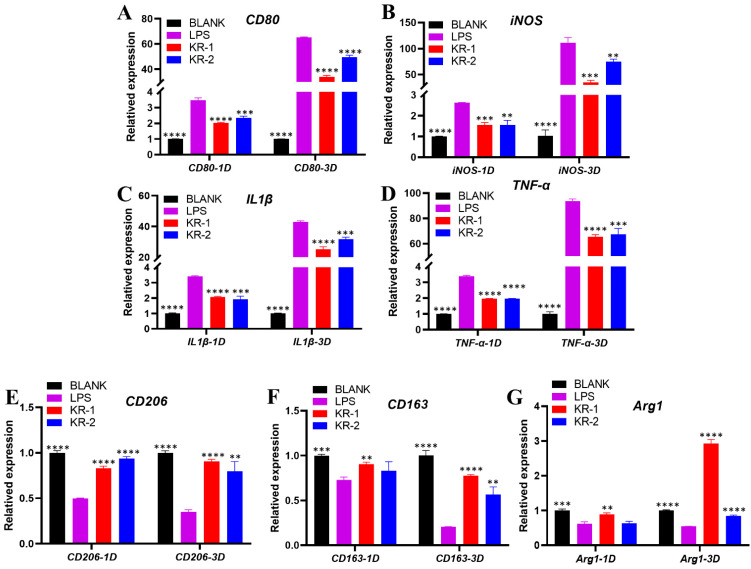
Under the conditions of LPS-induced inflammation, KR−1 and KR−2 were added to culture RAW 264.7 cells for 1 and 3 days, respectively, and the changes in the expressions of inflammation-related genes were detected by qPCR. (**A**–**D**) The levels of expression of pro-inflammatory-related genes. (**E**–**G**) The levels of anti-inflammatory-related genes. Note: data are presented as mean ±standard deviation (n = 3). * Indicates statistical significance between the LPS group and the other groups. (** *p* < 0.01, *** *p* < 0.001, and **** *p* < 0.0001, ANOVA).

**Figure 5 antibiotics-11-01482-f005:**
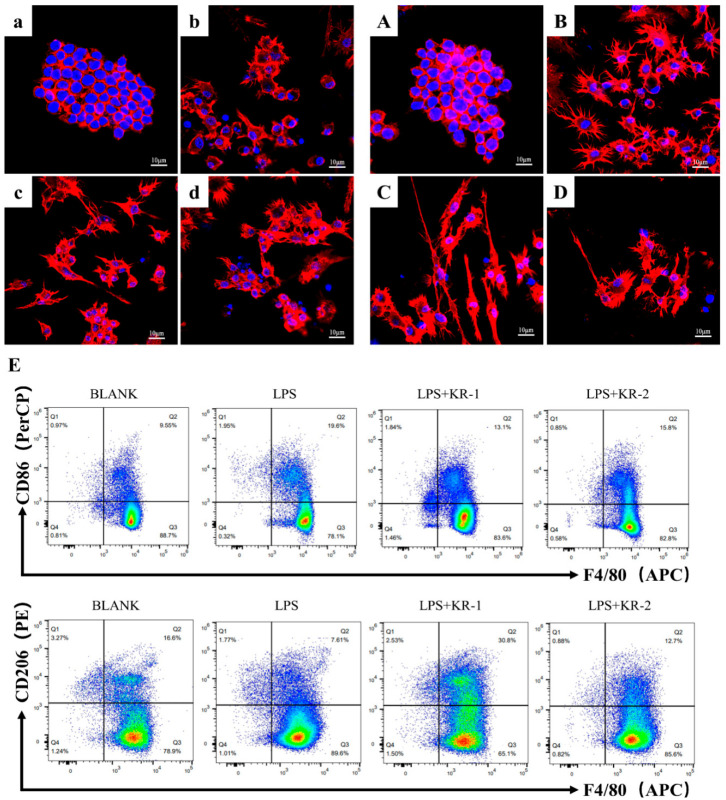
Regulation of macrophage phenotype in vitro. NOTE: CLSM images (**a**–**d**,**A**–**D**), lowercase letters represent images from day 1 and uppercase letters represent images from day 3. (**a**,**A**) BLANK control group. (**b**,**B**) LPS treatment group. (**c**,**C**) KR−1 and LPS were added at the same time. (**d**,**D**) KR−2 and LPS were added at the same time. (**E**) F4/80^+^/CD86^+^ and F4/80^+^/CD206^+^ cells detected by flow cytometry.

**Figure 6 antibiotics-11-01482-f006:**
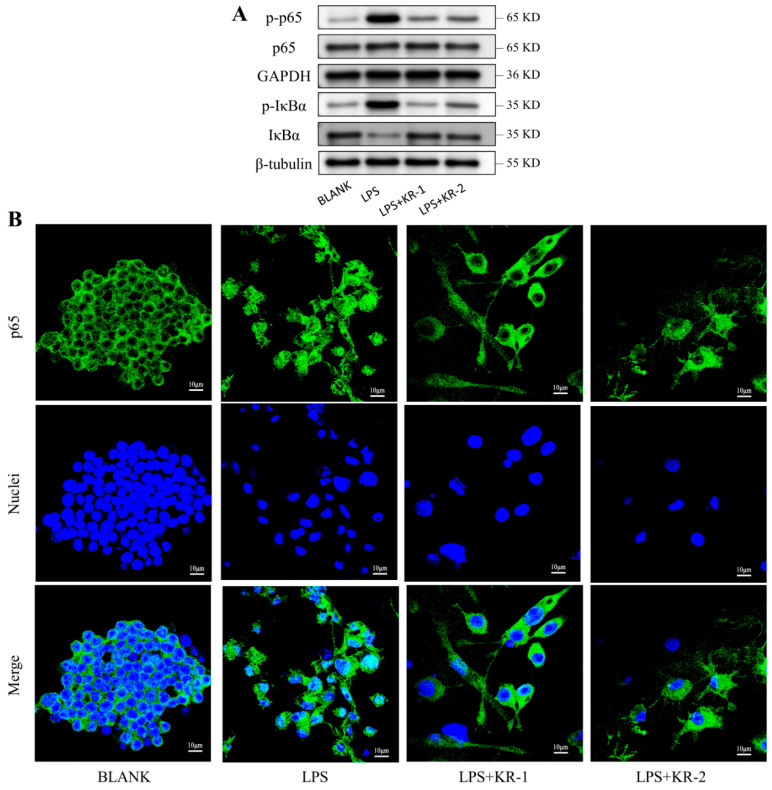
(**A**) Expression levels of NF-κB signaling pathway-related proteins: p65, p-p65, IκBα, and p-IκBα. (**B**) CLSM images of p65 in RAW264.7 cells after different treatments. Green (p65); blue (nuclei).

**Figure 7 antibiotics-11-01482-f007:**
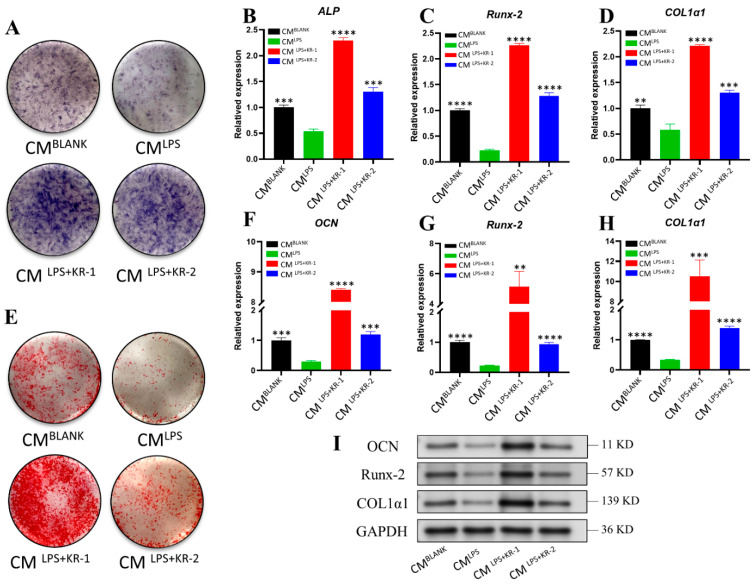
Regulation of osteogenesis of MC3T3-E1 cells by different groups of CM. (**A**,**E**) ALP staining (7 days) and ARS staining (14 days) of MC3T3-E1 cells cultured in an osteogenic medium supplemented with different sources of CM. (**B**–**D**) The expression levels of osteogenesis-related genes, ALP, Runx-2, and COL1α1, in different groups of CM cells cultured for 7 days. (**F**–**H**) The expression levels of osteogenesis-related genes, OCN, Runx-2, and COL1α1, in different groups of CM cells cultured for 14 days. (**I**) The expression levels of osteogenesis-related proteins, OCN, Runx-2, and COL1A1, at 14 days. Note: data are presented as mean ±standard deviation (n = 3). * Indicates statistical significance between the CMLPS and other groups. (** *p* < 0.01, *** *p* < 0.001, and **** *p* < 0.0001, ANOVA).

**Table 1 antibiotics-11-01482-t001:** In vitro susceptibility of *S. gordonii* and *F.nucleatum* to KR−1 and KR−2.

Bacteria	MIC (μg/mL)		MBC (μg/mL)	
	KR−1	KR−2	KR−1	KR−2
*S. gordonii*	125	50	250	150
*F. nucleatum*	100	50	200	100

**Table 2 antibiotics-11-01482-t002:** Primers and sequences used for qPCR in this study.

Gene (Mouse)	Forward Primer Sequence (5′-3′)	Reverse Primer Sequence (5′-3′)
CD80	CCTCAAGTTTCCATGTCCAAGGC	GAGGAGAGTTGTAACGGCAAGG
iNOS	GAGACAGGGAAGTCTGAAGCAC	CCAGCAGTAGTTGCTCCTCTTC
IL1β	TGGACCTTCCAGGATGAGGACA	GTTCATCTCGGAGCCTGTAGTG
TNF-α	GGTGCCTATGTCTCAGCCTCTT	GCCATAGAACTGATGAGAGGGAG
CD206	GTTCACCTGGAGTGATGGTTCTC	AGGACATGCCAGGGTCACCTTT
CD163	GGCTAGACGAAGTCATCTGCAC	CTTCGTTGGTCAGCCTCAGAGA
Arg-1	CATTGGCTTGCGAGACGTAGAC	GCTGAAGGTCTCTTCCATCACC
ALP	CCAGAAAGACACCTTGACTGTGG	TCTTGTCCGTGTCGCTCACCAT
Runx-2	CCTGAACTCTGCACCAAGTCCT	TCATCTGGCTCAGATAGGAGGG
COL1α1	CCTCAGGGTATTGCTGGACAAC	CAGAAGGACCTTGTTTGCCAGG
OCN	GCAATAAGGTAGTGAACAGACTCC	CCATAGATGCGTTTGTAGGCGG
GAPDH	CATCACTGCCACCCAGAAGACTG	ATGCCAGTGAGCTTCCCGTTCAG

Abbreviations: qPCR, quantitative real-time polymerase chain reaction; iNOS, inducible nitric oxide synthase; IL1β, interleukin-1β; TNF-α, tumor necrosis factor α; Arg-1, Arginine-1; ALP, alkaline phosphatase; Runx-2, runt-related transcription factor 2; COL1a1, collagen type I alpha I; OCN, osteocalcin; GAPDH, glyceraldehyde 3-phosphate dehydrogenase.

## Data Availability

All data included in this study are available upon request through contact with the corresponding author.

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
