# Peer review of "Study on Optimizing Novel Antimicrobial Peptides with Bifunctional Activity to Prevent and Treat Peri-Implant Disease"

_antibiotics, 2022, doi:10.3390/antibiotics11111482_

Round 1

Reviewer 1 Report

Presented studies relate to highly important topic concerning preventing and treatment of diseases around dental implants. The concept of research is well described and clear for the readers. However, there are few aspects I would like to highlight.

In the Section 2.1 the authors describe properties of studied peptides. Concerning the amino acid composition, I understand that the authors compare the number of hydrophilic groups present in KR-1 and KR-2 sequences, same as the net charge. However, I would recommend to avoid statements that KR-1 has relatively few hydrophilic groups and relatively low charge. 10 from 17 amino acids in KR-1 sequence are hydrophilic, which gives 59%, definitely not a few. The net charge of KR-1 is +10, only 1.4-fold lower than KR-2, hence I do not agree that its charge is low. I suggest the authors to change this part of the manuscript. Moreover, I do not understand, why in Figure 1 (A and B) there are Asn and His in the C-terminal part of both sequences? Because of those amino acids, there are 19 residues in both peptides, and, analyzing the sequences given in the Introduction, KR-1 and KR-2 have only 17. Moreover, I kindly ask for not using the phrase "C- or N-end" (like in lines 68 and 69). Please replace it with the phrases used in further sections of the manuscript.

Since the predictions of tertiary structure are not entirely reliable, I would highly recommend to perform NMR studies for both peptides to determine their full structures.

In the Section 2.3 I would suggest to clearly explain to the readers, what do presented pictures mean, without writing about specific colours (red and green). Meaning of colours is explained in the legend below Figure 2.

I insist of replacing the Figure 3B in present form with two separate diagrams (one for each peptide). Analyzing the x-axis, the scale is not acceptable. Please prepare two diagrams and place them next to each other. The results will be presented more clearly.

The phrase "peptide powder" (line 317) is not used in peptide studies. The authors could write just "peptide".

In the Section 4.1, there is no precise information about the concentrations of stock solutions and after dilution of both studied peptides. Please include those information.

In the Section 4.1, 4.2.2, 4.2.3 and 4.2.4, there is no information about the equipments used in described experiments (model, producer, country), while in further sections this information is included. Please complete those data.

Author Response

Point 1: In the Section 2.1 the authors describe properties of studied peptides. Concerning the amino acid composition, I understand that the authors compare the number of hydrophilic groups present in KR-1 and KR-2 sequences, same as the net charge. However, I would recommend to avoid statements that KR-1 has relatively few hydrophilic groups and relatively low charge. 10 from 17 amino acids in KR-1 sequence are hydrophilic, which gives 59%, definitely not a few. The net charge of KR-1 is +10, only 1.4-fold lower than KR-2, hence I do not agree that its charge is low. I suggest the authors to change this part of the manuscript. Moreover, I do not understand, why in Figure 1 (A and B) there are Asn and His in the C-terminal part of both sequences? Because of those amino acids, there are 19 residues in both peptides, and, analyzing the sequences given in the Introduction, KR-1 and KR-2 have only 17. Moreover, I kindly ask for not using the phrase "C- or N-end" (like in lines 68 and 69). Please replace it with the phrases used in further sections of the manuscript.

 Response 1: As reviewer said, the content of hydrophilic amino acids in KR-1 is not low. The original text simply compares the number of amino acids and net charges of the two, describing their relative amount. After re reading, we found that the original description is easy to cause misunderstanding to readers. According to reviewer’s suggestions, we will modify this part of the manuscript and only state the nature of the two peptides. We deleted the sentences in results part 2.1 about the hydrophilic and net charge of KR-1 and KR-2.

The Asn and His at the end of the sequence in Figure 1 (A and B) that review proposed was caused by our wrong use of the picture. When we predicted the basic properties of the sequence, we did not remove the residues at the end of the amino acid sequence in time at first, which led to the existence of 19 amino acids in the picture. However, the antibacterial peptides KR-1 and KR-2 that we designed only had 17 amino acids. After correction and prediction, the Figure was not updated in time, we are very sorry about that.

We have modified the phrase "C- or N-end" in the original text. Thank you for reviewer’s advice.

Point 2: Since the predictions of tertiary structure are not entirely reliable, I would highly recommend to perform NMR studies for both peptides to determine their full structures.

Response 2: We very much agree with you that studies such as nuclear magnetic resonance can indeed determine the complete structure of polypeptides. Our research group also discussed internally when formulating the experimental plan. Based on the previous experimental results, we found that there is no intuitive link between the structure detection results and the properties of short peptides. For the three-dimensional structure prediction software Alphafold selected in this paper, we selected it after consulting relevant research. Although the software is used to predict its three-dimensional structure, this software has been recognized by many scholars. In this study, the software was used to predict the amino acid sequence at least 5 times to ensure the accuracy of the experimental results. At the same time, the three-dimensional structure of amino acid sequence will change correspondingly when it performs its relative function. However, it is difficult to observe this phenomenon with our current experimental technology. For the above reasons, this study did not carry out NMR research.

Point 3: In the Section 2.3 I would suggest to clearly explain to the readers, what do presented pictures mean, without writing about specific colours (red and green). Meaning of colours is explained in the legend below Figure 2.

Response3: Thank you very much for reviewer’s suggestion. We have changed the original description. It is true that the changed description is more intuitive and easier to understand than before. The legend below Figure 2 had mentioned as “Note: Green represents live bacteria, red represents dead bacteria”.

Point 4: I insist of replacing the Figure 3B in present form with two separate diagrams (one for each peptide). Analyzing the x-axis, the scale is not acceptable. Please prepare two diagrams and place them next to each other. The results will be presented more clearly.

Response 4: Thank you very much for reviewer’s suggestion. We have processed Figure 3B according to reviewer’s suggestion to make the results clearer.

Point 5: The phrase "peptide powder" (line 317) is not used in peptide studies. The authors could write just "peptide".

Response 5: We are very sorry for the wrong use of "peptide powder". Thank you for pointing out our wrong use of words.

Point 6: In the Section 4.1, there is no precise information about the concentrations of stock solutions and after dilution of both studied peptides. Please include those information.

Response 6: Thank you very much for reviewer’s suggestion. The storage concentration of our polypeptide is 5mg/ml, which has been added in the original text. For the information of the experimental dilution concentration, the corresponding description is given in the corresponding section.

Point 7: In the Section 4.1, 4.2.2, 4.2.3 and 4.2.4, there is no information about the equipments used in described experiments (model, producer, country), while in further sections this information is included. Please complete those data.

Response 7: We are very sorry that we did not indicate the experimental equipment information in the original text. Thank you for pointing out our problems. We have made changes in the manuscript, indicating the equipment information used in the experiment.

Reviewer 2 Report

Yu et al. studied antimicroibal properties of the peptides. They have shown bacterial growth inhibition of these peptides. This is an interesting piece of work. But there are lack of some experimental data which needs to be addressed carefully before publication. I have several comments on this work which has been attached bellow

Comment 1: The author got the peptides from Dechi Biosciences 312 (Shanghai, China) and has given the physiochemical properties. But they have not explained how these peptides has been characterized. At least LC-MS spectra, MALDI spectra , Zeta potential curve is required for quality control. Also what is the control peptide they have used for this study (with respect to what peptide)

Comment 2: the folding of the peptides has been given but which software has been used for this study is not clear. Software name must be required. Also the control peptide folding properties must be given.

Comment 3: peptides MIC value has been determined but there is no control experiment (Table 1). With respect to which peptides author wants to claim these two peptides are good? Control peptides must be used and control experiment needs to be done.

Comment 4: Figure 2A what is the concentration of peptides for this OD determination? Figure 2B can author give the colocalization images for green and red? Also can author calculate the % of dead bacteria and live bacteria (kinetics)?

Comment 5: KR-1 showed no cytotoxicity at concentrations less than 256 μg/ml but it promoted cell proliferation better at a concentration of 64 μg/ml (Figure 3B). It is not clear Why these peptides promoted cell proliferation at 64 ug/ml concentration? More discussion needs to be added. Here also control peptide experiment need to be done.

Comment 6: Figure 4 concentration of peptides must be given.

Comment 7: Figure 5 It is showing the morphology changes of the RAW cell after peptide treatment. Is it good sign? Can author explain why these changes happening and what will be the effect good or bad? Also, will this peptides be nontoxic for other cells?

Comment 8: What is the stability of these peptides? Can author determine the t1/2 value of the peptide? How author treated the bacteria with the peptides? Is it one time treatment or peptides need to be added after a particular time interval?

Author Response

Point 1: The author got the peptides from Dechi Biosciences 312 (Shanghai, China) and has given the physiochemical properties. But they have not explained how these peptides has been characterized. At least LC-MS spectra, MALDI spectra , Zeta potential curve is required for quality control. Also what is the control peptide they have used for this study (with respect to what peptide)

 Response 1: Thank you very much for reviewer’s question. It is true that Figure 1 of the article only gives the physical and chemical properties of the polypeptide and the prediction of its spatial structure. However, for the polypeptide used, we used the LC-MS broad-spectrum for quality control after HPLC purification.

For the control peptide the reviewer mentioned, the reason why we did not set up a control group in the experimental design is that at present, it is mainly the improvement of KR-1 and KR-2. Each polypeptide has its own special performance, and it may not be possible to compare each performance with any polypeptide. In addition, in many cases, the improvement of polypeptides needs to adjust the spatial structure and properties, so a variety of polypeptides are needed as the control polypeptides. Such a setting first needs to understand a variety of polypeptides, which does not achieve the original purpose of this article. Therefore, we do not set the control polypeptide, but use the improved short peptide as the inter group control to intuitively adjust the polypeptide sequence to optimize performance.

Point 2: the folding of the peptides has been given but which software has been used for this study is not clear. Software name must be required. Also the control peptide folding properties must be given.

Response 2: Thank you for reviewer’s question. The name of the software to predict peptide folding is Alphafold, which is described in 4.1 of the original text. This software has been widely used in protein structure prediction research. There is no control peptide in this study, because the purpose of this study is to select the one with better comprehensive performance from the two improved antimicrobial peptides, so the two improved antimicrobial peptides are used as a cross reference to draw relevant conclusions.

Point 3: peptides MIC value has been determined but there is no control experiment (Table 1). With respect to which peptides author wants to claim these two peptides are good? Control peptides must be used and control experiment needs to be done.

 Response 3: The initial purpose of this study is to study the multiple properties of two new improved antibacterial peptides KR-1 and KR-2, and the related research of original antibacterial peptides is outside our research field, Therefore, there is no control group experiment for the pre modified polypeptide. Instead, the two improved polypeptides are compared between groups to select a better improved antibacterial peptide from the two. From this point of view, combined with the results of MIC, MBC, SEM and other experiments in this paper, the antibacterial performance of KR-2 is better than that of KR-1.

Point 4: Figure 2A what is the concentration of peptides for this OD determination? Figure 2B can author give the colocalization images for green and red? Also can author calculate the % of dead bacteria and live bacteria (kinetics)?

Response 4: As you mentioned in Figure 2A, the concentration of peptide when OD is measured is the MIC value of two peptides corresponding to different bacteria, for S. gordonii: KR-1 is 125 μg/ml, KR-2 is 50 μg/ml; for F. nucleatum: KR-1 is 100 μg/ml, KR-2 is 50 μg/ml.

As for the confocal image in Figure 2B the reviewer mentioned, I think the reviewer should mean the image of merge, and what we are showing is the captured merge image. For the blank control group, the number of live bacteria (green) is dominant, and the number of dead bacteria (red) is very small, but it can still be observed; For the experimental group, after the addition of antimicrobial peptides, the bacteria appear as a full screen of dead bacteria (red), and few live bacteria (green) are seen, which may also be related to the PBS flushing during our operation. However, the control group and the experimental group use the same PBS flushing steps, and the results obtained also clearly show that antimicrobial peptides have a significant killing effect on bacteria. We have conducted at least 3 repeated experiments, and selected the image with the best result for display.

The percentage of live/dead bacteria the reviewer proposed is a very good suggestion, which can more intuitively show the killing effect of antibacterial peptides on bacteria. However, according to the results obtained by CLSM in this study, after antimicrobial peptide treatment, living bacteria almost do not exist, so this paper does not calculate the percentage of living/dead bacteria.

Point 5: KR-1 showed no cytotoxicity at concentrations less than 256 μg/ml but it promoted cell proliferation better at a concentration of 64 μg/ml (Figure 3B). It is not clear Why these peptides promoted cell proliferation at 64 ug/ml concentration? More discussion needs to be added. Here also control peptide experiment need to be done.

 Response 5: Figure 3B shows the CCK-8 experiment for two kinds of peptides, which can reflect the cytotoxicity of the two new antibacterial peptides we designed. The experimental results show that KR-1 is at 64 μg/ml can promote cell proliferation. This result in this paper is mainly to show that KR-1 has better biocompatibility than KR-2. As for why the antibacterial peptide the reviewer mentioned promotes proliferation at this concentration, it may be necessary to consider the mechanism of action of antibacterial peptide on cells, but it is regrettable that the mechanism of action has not been studied in this study. Thank you for providing a good idea. The research team will discuss relevant issues and study its mechanism in the future.

As for the group of control peptides the reviewer gave that needs to be added, because the basic polypeptide is not consistent with the research field of this study, and this study mainly wants to compare the two improved antibacterial peptides, select the best improved antibacterial peptide from them, and compare them with each other, so there is no control group. I hope our answers can explain the reviewer’s questions.

Point 6: Figure 4 concentration of peptides must be given.

 Response 6: Thank you for raising this question. The concentration of antimicrobial peptides in Figure 4 and the subsequent experiments is described in Article 2.5, so the paragraphs after Article 2.5 do not repeat the experimental concentration. I'm sorry that our text is not clear enough.

Point 7: Figure 5 It is showing the morphology changes of the RAW cell after peptide treatment. Is it good sign? Can author explain why these changes happening and what will be the effect good or bad? Also, will this peptides be nontoxic for other cells?

Response 7: This experiment uses immunofluorescence staining to observe the morphological changes of RAW cells by staining the cytoskeleton, which is based on the research results of previous scholars. For this study, it is a good omen that macrophages tend to be elongated, which more intuitively shows that macrophages RAW264.7 tend to be M2 phenotype polarized.

Some researchers have previously proposed that macrophages RAW264.7 polarize to different phenotypes of M1 and M2, and cells will undergo different morphological changes. Relevant contents are discussed in Paragraph 5 of Discussion. Based on the current research results, this morphological change only indicates the polarization trend, and there is no difference between good and bad.

In this study, antibacterial peptides were used to act on RAW264.7 cells to observe their cytotoxicity. In addition, we also conducted relevant experiments on MC3T3-E1 cells and human bone marrow mesenchymal stem cells. It was found that the peptide concentration used in this experiment had no cytotoxicity on the above two cells, and whether there was cytotoxicity on other cells remained to be verified.

Point 8: What is the stability of these peptides? Can author determine the t1/2 value of the peptide? How author treated the bacteria with the peptides? Is it one time treatment or peptides need to be added after a particular time interval?

 Response 8: We are also very concerned about the stability of the peptide the reviewer proposed. Compared with proteins, short polypeptide chains have excellent stability. Based on the current experiments, in bacterial or cellular experiments, polypeptides can function for up to 3 days. Compared with the control group, the results of the experimental group have changed significantly, whether in terms of phenotype, gene or protein expression, this reflects that the short-term stability of polypeptide in solution is acceptable.

For the experiments that specifically reflect the stability of polypeptides and determine the t1/2 value of peptides, our experimental plan is to carry out the experiments in animals. The animal experiments of our research group are carried out in the animal experiment center outside the school. However, due to the repeated outbreaks in China, which hinder the development of relevant experiments in animals, no relevant results have been determined so far. We will study this question as part of the follow-up experiment plan.

The plan for treating bacteria with antimicrobial peptides in this study is as follows: In MIC, MBC determination and Biofilm Susceptibility Assay experiments, antimicrobial peptides are added to the bacterial suspension together with the culture medium for culture, while in CLSM and SEM experiments, bacteria are first cultured for a specific time, and then added peptide to the final concentration MIC value to continue to culture for 24h, and then fixed for observation, the relevant details are described in 4.2.1 to 4.2.4 of the manuscript.

Because the longest action period of polypeptides acting on bacteria in this study is 48h, the polypeptides in this study are all added once, and there is no subsequent addition.

Reviewer 3 Report

The manuscript demonstrates the optimization of antimicrobial peptides KR-1 and KR-2 based on 6K-F17 to apply to the tissue around the oral implant. Bioactivity profile of KR-1 and KR-2 by inhibiting the biofilm formation, bacterial cell-wall and cell membrane destruction, and activation of NF-κB signaling pathway supports the study's objectives. In my opinion, this manuscript can be accepted for publication in this journal if no other reviewer has any objections.

Author Response

Thank you very much for reviewing my manuscript in your busy schedule and for your affirmation of my work.

Round 2

Reviewer 2 Report

Authors have corrected the manuscript accordingly. This paper can be now published without any modification.